

# Analysis of factors influencing hookwire dislodgement in CT-guided hookwire localization: a retrospective study using variable importance analysis with a random forest model

Kiook Baek[1,2,3], Jin Young Kim[4] and Jung Hee Hong[4]

[1] Department of Occupational and Environmental Medicine, Dongguk University Gyeongju Hospital, Gyeongju, Republic of South Korea
[2] Department of Preventive Medicine, Dongguk University-Gyeongju, Gyeongju, Republic of South Korea
[3] Department of Medicine, Graduate School of Kyungpook National University, Kyungpook National University, Daegu, Republic of South Korea
[4] Department of Radiology, Keimyung University College of Medicine, Daegu, Dalseo-gu, Republic of South Korea

Corresponding author
Jung Hee Hong, keke4@naver.com

## ABSTRACT

**Background**. Video-assisted thoracoscopic surgery (VATS) is a minimally invasive and safe procedure. However, lung deflation during the operation causes anatomic landmark distortion, complicating small nodules detection. Computed tomography (CT)-guided hookwire localization promotes the success rates of VATS, but faces issues with hookwire dislodgement, potentially losing intraoperative tumor reference. This study was conducted to identify the relative importance ranking of potential factors influencing dislodgement in CT-guided hookwire localization.

**Methods**. This retrospective study reviewed 123 cases of CT-guided hookwire localization followed by VATS resection. Variables analyzed included sex, age, nodule size, emphysema, chest wall/muscle/total depth, distance from the nodule (DNP) or wire tip to the pleura (DWP), procedure time, nodule subtypes, multiple localization, post-procedural hemorrhage, pneumothorax, nodule penetration, and time intervals between completion of procedure to initiation of surgery (PS interval). Variables were compared using chi-square tests or Mann-Whitney tests. A random forest model, enhanced with the Synthetic Minority Over-sampling Technique (SMOTE) for oversampling, was employed to determine the relative importance of each variable. The relative importance of variables was presented using the mean decrease Gini and mean decrease accuracy metrics. For sensitivity analysis, relative variable importance was analyzed using extreme gradient boosting (XGBoost) model, and the relative importance of variables was presented using the gain metric.

**Results**. Among the 123 cases, dislodgement occurred in 15. In univariable analysis, only the PS interval was statistically significant ($134.1 \pm 73.1$ *vs.* $104.1 \pm 46.1$ minutes in dislodgement or non-dislodgement, $p = 0.031$). The random forest and XGBoost model identified the top five important variables as the PS interval, DWP, DNP, total depth, and age. The top five factors demonstrated a distinct difference when compared to the other factors.

**Conclusions**. The study identified the PS interval as the most critical factor in hookwire dislodgement, along with DNP, DWP, total depth, and age. These results identified the presence of modifiable factors within the hospital and can assist practitioners and surgeons in recognizing the dislodgement risk of procedures based on various patient factors.

# INTRODUCTION

With the recent increased in the use of chest computed tomography (CT), the detection of solitary pulmonary nodules has significantly risen, presenting a diagnostic challenge. Small-sized pulmonary nodules are particularly difficult to differentiate as benign or malignant solely based on CT scans, necessitating tissue confirmation for further treatment planning. Video-assisted thoracoscopic surgery (VATS) is a minimally invasive procedure that has proven to be both safe and technically feasible (*Bernard & The Thorax Group, 1996*; *Congregado et al., 2008*; *Suzuki et al., 1999*). It offers less surgical trauma, faster postoperative recovery, and fewer complications compared to thoracotomy (*Mack et al., 1993*). However, lung deflation during the operation inevitably causes anatomic landmark distortion, complicating the detection of small parenchymal nodules or ground-glass nodules (GGNs), which are often neither visible nor palpable (*Kim, 2022*; *Lenglinger, Schwarz & Artmann, 1994*).

Techniques for preoperative localization have been implemented to increase the success rates of VATS and reduce the need for unplanned thoracotomy (*Chella et al., 2000*; *Ichinose et al., 2013*; *Kawanaka et al., 2009*; *Saito et al., 2002*; *Suzuki et al., 1999*). Despite the availability of various techniques for preoperative localization of pulmonary nodules, CT-guided hookwire localization remains one of the oldest and most commonly used methods, with success rate of 93.6–97.6% (*Ciriaco et al., 2004*; *Lin & Chen, 2016*; *Mack et al., 1992*). However, hookwire dislodgement is a significant issue with this method, potentially leading to the loss of intraoperative reference for the tumor (*Chen et al., 2011*; *Hanauer et al., 2016*; *Seo et al., 2012*). Dislodgement not only poses a challenge during surgery but can also necessitate additional procedures.

Concerns regarding the dislodgement associated with hookwire localization have prompted extensive research and the use of alternative localization methods, such as methylene blue, metallic coils, lipiodol and indocyanine green (*Hsu et al., 2024*; *Park et al., 2020*; *Wang et al., 2023*; *Zhang et al., 2022*). However, these methods have their own limitations. Methylene blue is prone to fading, indocyanine green is eliminated through bile secretion, and dyes may become invisible during surgery when injected into thick central lung parenchyma. Additionally, metallic coil insertion or lipiodol injection requires intraoperative fluoroscopic assistance to confirm localization, which can complicate the surgical procedure (*Hsu et al., 2024*). Thus, despite the risk of dislodgement and the

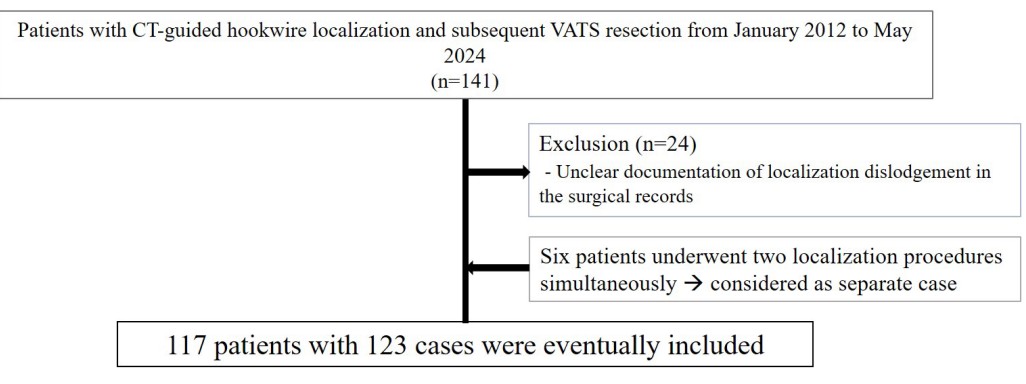

**Figure 1  Case inclusion flow chart.**

development of alternative localization tools, CT-guided hookwire localization remains widely used due to its precision and reliability.

Despite the importance of this issue, there is a paucity of empirical studies identifying the factors crucial for hookwire nodule localization. Recently, various machine learning techniques have been applied in engineering (*Akırmak & Altan, 2023*) and medical fields (*Esmaily et al., 2018*; *Ooka et al., 2021*), and these methods are increasingly used to analyze and predict the impact of different factors on outcomes. In this study, we aim to explore the influence of various factors on the occurrence of dislodgement during the procedure. These factors include those previously identified as risk factors in similar procedures, as well as clinical and anatomical variables presumed to impact the procedure. To achieve this, we utilized machine learning techniques to investigate the relative importance of these factors.

# MATERIALS AND METHODS

This retrospective study was approved by the institutional review board of Dongsan Medical Center (IRB number DSMC IRB 2024-05-004); the requirement for patients' informed consent was waived.

## Study design and patients

We reviewed medical records from January 2012 to May 2024 at one tertiary institution to investigate cases of CT-guided hookwire localization and subsequent VATS resection for lung nodules. We checked where the surgical records documented the success or failure of localization, and cases without documented localization outcomes were excluded from the study. Cases in which localization was performed on two nodules during a single surgery, followed by VATS resection for both nodules, were also included. In such instances, each nodule was considered as a separate case for the purposes of this study. For example, if a single patient underwent localization for two nodules, it was classified and analyzed as two separate cases (Fig. 1).

### CT-guided localization technique

CT-guided localization is a technique performed in a CT room, where imaging is used to identify the position of the target nodule, and a hookwire is inserted adjacent to the nodule under CT guidance. During the localization procedure, a limited scan range of the chest was covered to visualize the target nodule. All localization was conducted with 64- or 128- slice multidetector CT scanners (SOMATOM definition Edge and SOMATOM sensation 64; Siemens Medical Solutions, Forchheim, Germany). The scan parameters were as follows: 100 mA, 100 kVp and pitch of 1–1.2. The scan data were reformatted with a 3.0 mm section thickness for the transverse images. A 7.5 cm or 10 cm length with 20 gauge size localization hookwire (Argon Localization Needle, Argon Medical, Plano, TX, USA) was used for nodule localization. The CT images with limited scan coverage are reviewed to determine the needle puncture site. Subsequently, local anesthesia is administered to the skin around the selected puncture site. The cannula needle housing the hookwire is then gradually inserted through the chest wall and into the lung parenchyma under sequential CT guidance. Once the desired position is reached, the outer cannula needle is withdrawn, releasing the hookwire's horn, which anchors the wire securely in place. Finally, a CT image is obtained with the hookwire secured in its final position, allowing the thoracic surgeon to understand the relationship between the insertion site and the nodule before surgery, ensuring optimal surgical planning and execution.

### Potential factor selection

Dislodgement occurring during the procedure was defined as the outcome. Potential predictor variables were selected considering procedure-related characteristics, radiologic features of patient findings, general characteristics, and in-hospital management factors. Procedure-related characteristics included distance from the nodule to the pleura (DNP), distance from the wire tip and the pleura (DWP), total depth, muscle depth, chest wall depth, multiple localization, presence of post-procedure pulmonary hemorrhage, presence of post-procedure pneumothorax, nodule penetration and procedure time. For an in-hospital management characteristic, PS interval was selected. General patient characteristics included age and sex, while radiologic features related to the patient's lesion and lungs comprised nodule size, nodule subtype, and emphysema.

### Procedure-related characteristics

CT images obtained during CT guided localization were reviewed by a thoracic radiologist (J.H.H.) who was blinded whether the case was of dislodgement or not. The maximum diameter of the nodule from skin (total depth), the thickness of the chest wall and muscle layer (chest wall depth and muscle depth) traversed by the localization hookwire, DWP, DNP were measured using electronic calipers (millimeters) (Fig. 2). Additionally, we checked whether the localization hookwire penetrated the nodule (nodule penetration). Post procedural pneumothorax and pulmonary hemorrhage were recorded. Pulmonary hemorrhage was defined as new consolidation or GGNs on post-procedural images and was recorded as presence of hemorrhage, including needle tract hemorrhage less than two cm in width (*Tai et al., 2016*). We also assessed cases performing multiple localizations during

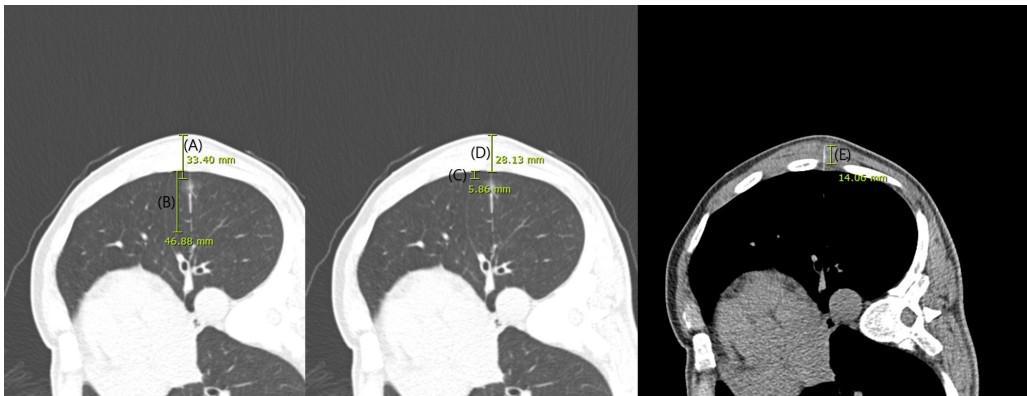

**Figure 2** **CT images obtained during localization.** The maximum diameter of the nodule from skin (total depth; (A), distance from the wire tip to the pleura (B), distance from the nodule to the pleura (C), the thickness of the chest wall (D) and muscle layer (E) traversed by the localization hookwire were measured using electronic calipers (millimeters).

a single localization procedure (multiple localization). The initiation and completion time of the CT-guided localization procedure was reviewed to evaluate procedure time.

## Patient-related characteristics

Sex and age were examined my medical record. Preoperative CT scans were reviewed by a thoracic radiologist (J.Y.K). The radiologist evaluated the nodule size and subtypes and location of the nodule. The nodule subtype was classified into three categories: pure GGNs, part-solid nodules (PSNs), and solid nodules (*Sun et al., 2024*). The degrees of emphysema (0, none; 1, trace or mild; 2, moderate; 3, confluent; 4, advanced destructive) was defined based on the CT-based Visual Classification of Emphysema, which has been used in various studies (*Lynch et al., 2018*).

## In-hospital management related characteristics

The initiation of the CT-guided localization procedure, and the initiation times of the surgery were reviewed to evaluate PS interval.

## Outcome

The presence of hookwire dislodgement in CT-guided hookwire localization was determined by reviewing operation notes (surgical record). The surgeon documented whether the wire localization was well positioned in the operation field. We retrospectively reviewed the operation notes, and cases where the documentation did not clearly state were excluded from the analysis ($n = 24$).

## Statistical methods

The groups were classified based on the presence or absence of dislodgement. The characteristics of each group were then summarized by presenting the mean and standard deviation. We conducted Mann–Whitney test for comparing continuous variables among both groups, due to the small number of dislodgement cases ($n = 15$). Chi-squared

test with Yates's correction was performed for comparing categorical variables. PS interval (minutes), DNP (mm), DWP (mm), total depth (mm), age (years), procedure time (minutes), nodule size (mm), muscle depth (mm), and chest wall depth (mm) were analyzed as continuous variables. Nodule subtype (pure GGNs, PSNs, and solid nodules), sex (male/female), pulmonary hemorrhage (present/absent), emphysema (grade 0/1/2/3/4), multiple localization (yes/no), nodule penetration (yes/no) and pneumothorax (present/absent) were analyzed as categorical variables.

We aimed to analyze relative variable importance and rank the potential variables. Given the prevalence of categorical variables in the dataset, a random forest model was employed as an analytical method to assess the importance of each variable's influence on the outcome variable. Random forest is an ensemble learning technique that enhances predictive performance by aggregating multiple decision trees. Each tree is trained on different samples, which increases the model's robustness. The model is trained by repeatedly sampling a subset of the data with replacement (bootstrapping) to create multiple decision trees. Each tree in the forest is built from a different bootstrap sample, which typically includes about two-thirds of the original data. The remaining one-third of the data, known as out-of-bag (OOB) samples, which are not used in the training of each tree, allow for an assessment of the model's accuracy and stability without the need for separate validation data (*Breiman, 2001*). The number of trees was set to 500, and the number of variables tried at each split was set to 4, which is the square root of the total number of predictors. The minimum size of terminal nodes was set to 1. The splitting criterion used for node division was the Gini Index. The Gini Index, which measures node impurity, was calculated as $Gini = 1 - \sum p_i^2$, where $p_i$ is the proportion of samples in class $i$. For each node, the model selects the variable and splitting point that minimize the weighted Gini Index of the child nodes. Tree depth, representing the number of nodes from the root to the deepest leaf node, was not explicitly restricted during model construction, allowing the trees to grow until terminal nodes contained the minimum number of samples (one sample per node). Tree depth was presented to understand the complexity of the individual trees, and the average, maximum, and minimum depths were presented.

Due to the significant imbalance in the data distribution, with group sizes of 15 and 108, it was deemed inappropriate to classify the data using a random forest model directly. Thus, the Synthetic Minority Over-sampling Technique (SMOTE) method was applied to address this issue. SMOTE addresses data imbalance by synthesizing new samples for the minority class and adjusting the majority class through oversampling and undersampling (or random sampling), respectively. In SMOTE, oversampling of the minority class (the group with dislodgement) was performed using the K-nearest neighbors (KNN) algorithm. Specifically, KNN was employed to identify the K-nearest neighbors for each minority class data point, and new synthetic data points were generated by creating linear combinations of these neighbors and the existing data points (*RColorBrewer & Liaw, 2018*). In our implementation, the number of nearest neighbors was set to 5. For each minority class instance, synthetic samples were created by interpolating between the instance and five of its nearest neighbors. The minority class data was increased by 600%, and for the majority

class (the group without dislodgement), random sampling with replacement was conducted to match 150% of the newly oversampled minority class.

The results of the random forest model were described using OOB (out-of-bag) data, focusing on the sensitivity, specificity, accuracy of the predicted values compared to the actual values, and the $P$-value (for accuracy > no information rate). The validation of the random forest model, trained on the SMOTE-oversampled dataset, was performed using 10-fold cross-validation, and the results were included in the analysis. Additionally, the model was applied to the original dataset without SMOTE, and the sensitivity, specificity, accuracy, and $P$-value were also described. The importance of variables, as results of random forest model, were presented using mean decrease accuracy (MDA) and mean decrease Gini (MDG). These metrics provide insights into the relative importance of each predictor in the model. MDA evaluates variable importance in random forest by measuring the drop in model accuracy when a variable's values are randomly permuted. Larger decreases indicate higher importance, and MDA is calculated using OOB data. MDG measures the importance of each variable by calculating the decrease in the Gini impurity when the variable is used in the tree splits (*Han, Guo & Yu, 2016*).

As part of the sensitivity analysis, the variable importance ranking was re-evaluated using the eXtreme gradient boosting (XGBoost) method. The minimum sum of instance weights required in a child was set to 1. The maximum depth of the decision trees set to 6. The learning rate was set to 0.01. The number of boosting iterations was set to maximum 1,000, with early stopping enabled. The subsample rate was set to 0.8. Feature importance was evaluated using the Gain metric, which was calculated based on the XGBoost model. The importance of each variable was determined by assessing its contribution to the reduction in the model's loss function during the tree-building process. The Gain represents the average improvement in model performance, where a higher Gain value indicates that the feature plays a more significant role in reducing prediction errors. Due to the nature of XGBoost, which accepts input values in the form of a numeric matrix, variables with three or fewer categories were converted into dummies. For the three-category variable, nodule subtype, the reference variable ("00" coding) was set as pure GGN. Variables with more than three categories, such as emphysema grade, were encoded using one-hot encoding.

R project 4.4.0 (https://r-project.org) was used for statistical analysis. Package "randomForest" was used for random forest modelling, "caret" for k-fold validation of random forest, "DMwR" for SMOTE, "xgboost" for XGBoost modeling.

# RESULTS

## Study participants

A total of 141 patients were identified, of which 24 were excluded due to unclear documentation of localization dislodgement in the surgical records. Six patients underwent two localization procedures simultaneously. A total of 123 procedures were performed on 117 patients.

## General characteristics

The general characteristics of the participants are presented in Table 1. Among 123 included cases, 15 cases (12.2%; M:F, 7:8) showed wire dislodgement. The mean age was 67.2 ± 14.0 years in the dislodgement group and 66.0 ± 10.7 years in the non-dislodgement group. Of these dislodged 15 cases, eight nodules were located in the lower lobes (right lower lobe, 5; left lower lobe, 3), and seven were in the upper or middle lobes (right upper lobe, 5; right middle lobe, 1; left upper lobe, 1). The mean nodule size was 11.7 mm (range, five mm to 28 mm). Seven cases were solid nodules, seven were PSNs and one case was GGN. Among these, seven nodules were penetrated with the hookwire, while eight were not. Age, sex, emphysema degree, nodule subtypes (solid, GGNs or PSNs) and nodule size were not statistically significant differences between the dislodgement and non-dislodgement groups (Table 1). Despite the dislodgement, resections of all 15 cases were successfully performed, although two cases were converted to lobectomy instead of wedge resection. Both cases were confirmed as lung adenocarcinoma, and lobectomy was the appropriate treatment for them.

During the localization procedure, parenchymal hemorrhage was observed in 53 cases (43.0%). Pneumothorax occurred in 61 cases (49.0%). In the dislodgement group, five cases (33.3%) of pneumothorax and nine cases (60.0%) of parenchymal hemorrhage occurred. In the non-dislodgement group, 56 cases (51.9%) of pneumothorax and 44 cases (40.7%) of parenchymal hemorrhage occurred. No statistical significant difference were observed in both groups according to post procedural complications. None of the patients required specific intervention associated for post-procedural complications.

The PS interval was significantly different between dislodgement and non-dislodgement group, with 134.1 ± 73.1 min in the dislodgement group and 104.1 ± 46.1 min in the non-dislodgement group ($p = 0.031$). However, total depth (58.4 ± 13.6 mm and 60.2 ± 19.2 mm, respectively), DNP (16.3 ± 11.2 mm and 16.1 ± 14.3 mm, respectively), DWP (24.6 ± 15.5 mm and 31.4 ± 14.0 mm, respectively), procedure time (19.3 ± 17.5 min and 15.3 ± 7.1 min, respectively), chest wall depth (41.7 ± 12.0 mm and 41.1 ± 11.9 mm, respectively), muscle depth (18.9 ± 11.2 mm and 19.0 ± 11.0 mm, respectively) were not significantly different between the two groups (Table 1). Presence of nodule penetration and multiple localization were also not statistically different (dislodgement *versus* non-dislodgement, penetrated, seven cases, 46.7% *versus* 55 cases, 50.9%; presence of multiple localization, four cases, 26.7% *versus* eight cases, 7.4%).

## The performance of the random forest model

Using the SMOTE for oversampling, the dataset was constructed with 105 cases in the dislodgement group (15 original cases and 90 synthetic cases) and 135 cases in the non-dislodgement group (random sampling with replacement from 108 original cases). The characteristics of the dataset after performing SMOTE are provided in (Table S1).

The tree depths ranged from a minimum of seven to a maximum of 16, with an average depth of 10.1. The error rate performed on OOB data was 5.8%. Among the 240 total cases including synthetic data, the model correctly predicted dislodgement occurrence in 96 cases (true positives) and correctly identified non-dislodgement in 130 cases (true

**Table 1  Clinical and radiologic characteristics of 123 patients with CT-guided hookwire localization, including subgroup analysis of dislodgement (n = 15) and non-dislodgement (n = 108) groups.**

| | Dislodgement group N = 15 | Non-dislodgement group N = 108 | Total N = 123 | p-value* |
|---|---|---|---|---|
| Sex[b] | | | | 0.615 |
| 1. Male | 7 (46.7%) | 62 (57.4%) | 69 (56.1%) | |
| 2. Female | 8 (53.3%) | 46 (42.6%) | 54 (43.9%) | |
| Age (years)[a] | 67.2 ± 14.0 | 66.0 ± 10.7 | 66.1 ± 11.1 | 0.341 |
| Emphysema[b] | | | | 0.734 |
| 1. None | 9 (60.0%) | 56 (51.9%) | 65 (52.8%) | |
| 2. Trace or mild | 6 (40.0%) | 44 (40.7%) | 50 (40.7%) | |
| 3. Moderate | 0 (0.0%) | 5 (4.6%) | 5 (4.1%) | |
| 4. Confluent | 0 (0.0%) | 3 (2.8%) | 3 (2.4%) | |
| 5. Advanced destructive | 0 (0.0%) | 0 (0.0%) | 0 (0.0%) | |
| Nodule subtype[b] | | | | 0.169 |
| 1. Solid | 7 (46.7%) | 61 (56.5%) | 68 (55.3%) | |
| 2. PSNs | 7 (46.7%) | 27 (25.0%) | 34 (27.6%) | |
| 3. GGNs | 1 (6.7%) | 20 (18.5%) | 21 (17.1%) | |
| Nodule size (mm)[a] | 11.7 ± 7.2 | 11.7 ± 6.4 | 11.7 ± 6.5 | 0.774 |
| Total depth (mm)[a] | 58.4 ± 13.6 | 60.2 ± 19.2 | 60.0 ± 18.6 | 0.914 |
| Chest wall depth (mm)[a] | 41.7 ± 12.0 | 41.1 ± 11.9 | 41.2 ± 11.8 | 0.657 |
| Muscle depth (mm)[a] | 18.9 ± 11.2 | 19.0 ± 11.0 | 19.0 ± 11.0 | 0.963 |
| Distance between nodule to the pleura (mm)[a] | 16.3 ± 11.2 | 16.1 ± 14.3 | 16.1 ± 13.9 | 0.618 |
| Distance between wire tip to pleura (mm)[a] | 24.6 ± 15.5 | 31.4 ± 14.0 | 30.6 ± 14.3 | 0.074 |
| Presence of nodule penetration[b] | | | | 0.973 |
| 1. No | 8 (53.3%) | 53 (49.1%) | 61 (49.6%) | |
| 2. Penetrated | 7 (46.7%) | 55 (50.9%) | 62 (50.4%) | |
| Procedure time (minutes)[a] | 19.3 ± 17.5 | 15.3 ± 7.1 | 15.8 ± 9.0 | 0.702 |
| PS interval (minutes)[a] | 134.1 ± 73.1 | 104.1 ± 46.1 | 107.8 ± 50.8 | **0.031** |
| Pneumothorax[b] | | | | 0.285 |
| 1. None | 10 (66.7%) | 52 (48.1%) | 62 (50.4%) | |
| 2. Yes | 5 (33.3%) | 56 (51.9%) | 61 (49.6%) | |
| Hemorrhage[b] | | | | 0.257 |
| 1. None | 6 (40.0%) | 64 (59.3%) | 70 (56.9%) | |
| 2. Yes | 9 (60.0%) | 44 (40.7%) | 53 (43.1%) | |
| Presence of multiple localization[b] | | | | 0.059 |
| 1. No | 11 (73.3%) | 100 (92.6%) | 111 (90.2%) | |
| 2. Yes | 4 (26.7%) | 8 (7.4%) | 12 (9.8%) | |

**Notes.**
PS interval, the interval between the completion of the localization procedure and the initiation of surgery.
[a] Data are mean ± standard deviation.
[b] Data are number of patients, with percentages in parentheses.
*Result in bold indicates a significant finding.
P-value was calculated with Mann–Whitney test.

negatives). It failed to predict dislodgement in nine cases (false negatives) and incorrectly predicted dislodgement in five cases (false positives), resulting in a sensitivity of 0.91 (95% CI [0.92–0.99]) and a specificity of 0.96 (95% CI [0.84–0.96]). The classification

accuracy was 0.94 (95% CI [0.90–0.98]), and the *P*-value (for accuracy > no information rate) was <0.0001, indicating significance. The negative predictive value was 0.93 (95% CI [0.88–0.97]), and the positive predictive value was 0.95 (95% CI [0.89–0.98]). The F1 score was 0.95, and the Kappa coefficient was 0.88. The results of the 10-fold cross-validation performed on the oversampled dataset showed an accuracy of 0.95 (95% CI [0.92–0.98]) and a Kappa value of 0.90 (95% CI [0.84–0.96]). Considering the model was built with synthetic data, another validation was performed on the original dataset (15 dislodgement cases and 108 non-dislodgement cases). Of the cases predicted to be positive (group where dislodgement occurred), 22 cases were identified, with 15 being actual cases of dislodgement and seven being false positives. In the group predicted to be negative (group where dislodgement did not occur), 101 cases were identified, with no false negatives. The classification accuracy was 0.94 (95% CI [0.89–0.98]), and the *P*-value (for accuracy > no information rate) was 0.01, indicating significance. Assuming dislodgement cases as positive, the model demonstrated a specificity of 0.93 (95% CI [0.87–0.97]) and a sensitivity of 1.00 (95% CI [0.78–1.00]). The negative predictive value was 1.00 (95% CI [0.96–1.00]), while the positive predictive value was 0.68 (95% CI [0.45–0.86]). The Kappa coefficient was 0.78.

## Analysis of variable importance order using a random forest model

In the random forest model, variable importance was assessed using mean decrease accuracy and mean decrease Gini criteria. The most important variable was PS interval, followed by DWP, DNP, total depth, and age. A sharp decline in importance was observed between the top five variables and the sixth variable. Variables such as chest wall depth, procedure time, muscle depth, nodule size, and nodule subtype formed a subsequent group, with another decline in importance observed at the 11th variable. The bottom five variables for both criteria were multiple localization, sex, hemorrhage, emphysema, pneumothorax, and presence of nodule penetration (Fig. 3).

## Analysis of variable importance order using a XGBoost model

The sensitivity analysis using XGBoost for variable importance ranking also identified PS interval as the most significant variable explaining dislodgement. Following this, age, DWP, total depth, and DNP were ranked in descending order of importance. While there were differences in the specific order, the top five variables identified were consistent with those from the random forest model. Variables ranked after the top five included nodule subtype (solid nodule), sex, and wall depth in descending order. The graph illustrating variable importance derived from the XGBoost analysis is provided as a Fig. 4.

## DISCUSSION

This study identified factors influencing the dislodgement of CT-guided hookwire localization. Univariate analysis showed that PS interval was the only significant factor for dislodgement. We utilized various multivariate analysis and machine learning techniques such as SMOTE and random forest for deeper analysis. In both the random forest model and XGBoost model, five factors—DNP, DWP, total depth, and age—showed relatively

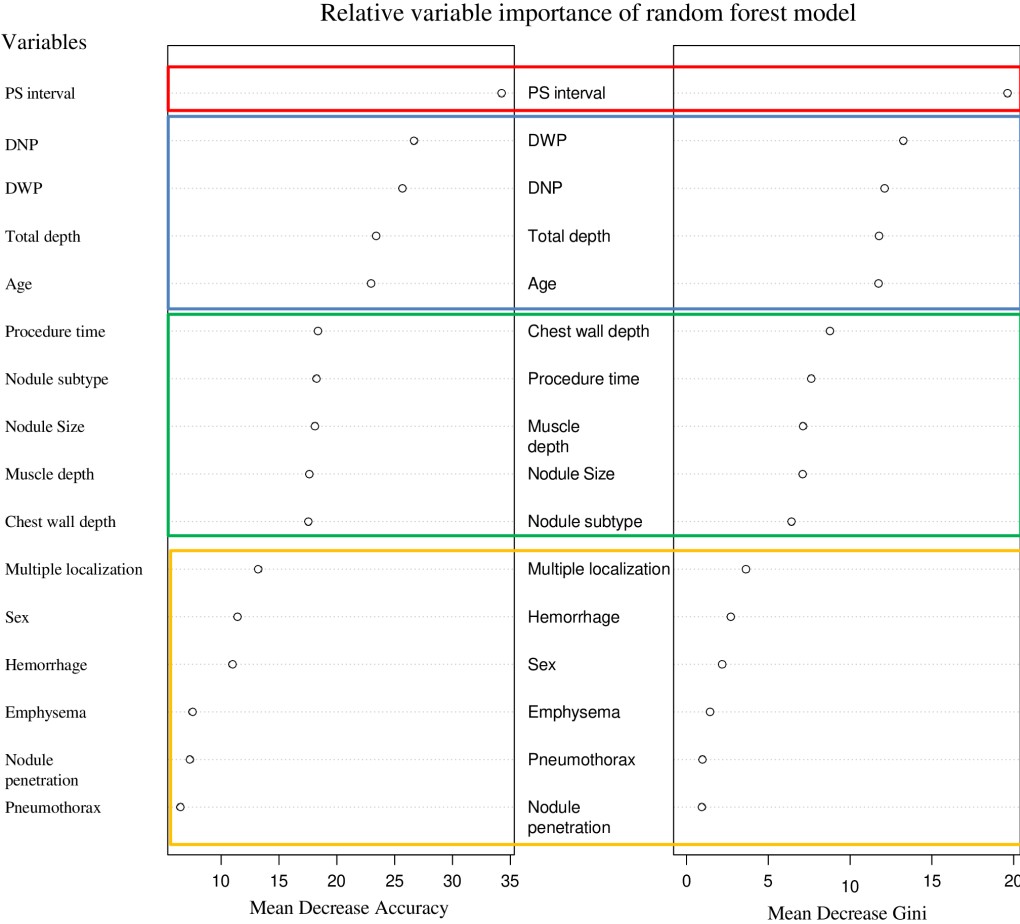

**Figure 3 The variable importance rankings derived from the random forest model.** These are represented using two metrics: mean decrease accuracy and mean decrease Gini. Mean decrease accuracy (MDA) measures the importance of a variable by evaluating the decrease in model accuracy when the values of that variable are randomly shuffled. Mean decrease Gini (MDG) assesses variable importance based on the reduction in Gini impurity across all trees in the forest, reflecting how often a variable is used for splitting the data. These values provide a way to compare variables, with higher values indicating greater contribution to model performance, but they do not have absolute units. It can be observed that the rankings of the top 1, 2–4, 5–9, and 10–16 groups are consistent across both indicators. Note: PS interval, the interval between the completion of the localization procedure and the initiation of surgery; DNP, distance from the nodule to the pleura; DWP, distance from the wire tip to the pleura.

high priority in predicting dislodgement. Factors such as the occurrence of pneumothorax during the procedure, presence of nodule penetration, and emphysema demonstrated the lowest importance in classification. Notably, the top five variables yielded identical results in the sensitivity analysis conducted using XGBoost, further reinforcing their significance.

Our study builds on previous research assessing successful CT-guided hookwire localization and provides further evidence that the PS interval is a critical factor influencing hookwire dislodgement. Earlier studies identified DWP as the most important determinant for successful nodule localization, with one study reporting a single dislodgement out of 17 cases when the interval between the procedure and surgery was delayed by six hours, and

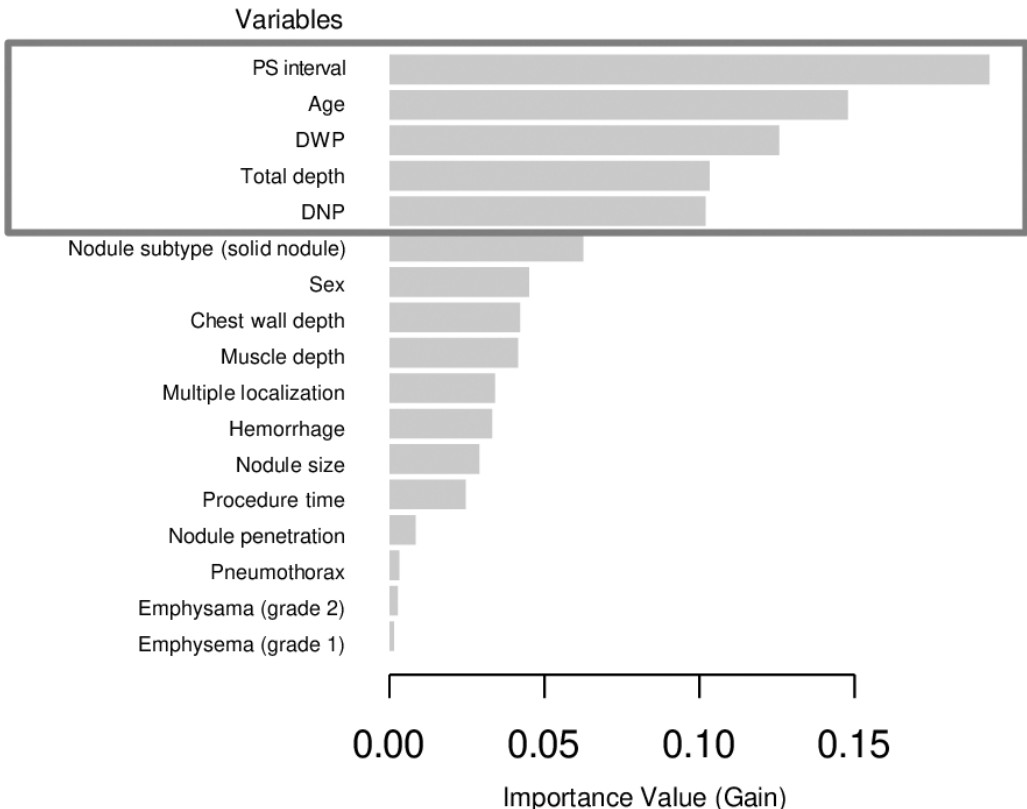

**Figure 4 The variable importance rankings derived from the extreme gradient boosting (XGBoost) model.** The importance was calculated using the gain metric, which measures the contribution of each variable to the model's performance improvement at each decision split. Higher gain values indicate greater importance of the corresponding variable in the model. Notably, the top five variables identified are consistent with the results from the random forest model, Note: PS interval, the interval between the completion of the localization procedure and the initiation of surgery; DNP, distance from the nodule to the pleura; DWP, distance from the wire tip to the pleura.

another highlighting dislodgements during patient transfer to the surgical suite (*Dendo et al., 2002*; *Seo et al., 2012*; *Shah et al., 1993*; *Zhao et al., 2022*). Previous analysis has focused on factors such as age, sex, nodule size, subtype, location, DWP, DNP, chest wall depth, and the angle between the wire and pleura, consistently identifying DWP as the primary independent factor for successful localization (*Seo et al., 2012*). In our study, we similarly measured DWP, DNP, chest wall depth, total depth, and muscle depth, while incorporating additional variables such as nodule subtype, size, and post-procedural complications. Importantly, we introduced a novel variable, the PS interval, which reflects the time between localization and surgery, to address concerns about the risk of dislodgement during patient transfer. Our findings revealed that while DWP remains a key factor consistent with prior research, the PS interval emerged as the most predictive factor for dislodgement (*Seo et al., 2012*). This suggests that delays in surgery may amplify risks, as

longer intervals necessitate extended patient transfer, increasing the likelihood of hookwire displacement. By identifying the PS interval as a dominant factor, our study emphasizes the importance of not only radiologists' and surgeons' technical expertise and patient-specific considerations but also timely surgical procedures following localization to minimize the risk of dislodgement and improve procedural success.

Other relatively important factors identified include the DNP, DWP, total depth, and age. While the DWP or total depth, related to the radiologist's technique, were one of the relatively important factors, others such as procedure time, chest wall depth, and muscle depth were relatively less important. The presence of hemorrhage or pneumothorax during the procedure and whether the nodule was penetrated showed very low importance. Chest wall depth was not identified as a key factor in dislodgement, aligning with previous findings that emphasize the importance of needle insertion depth over total chest wall thickness (*Seo et al., 2012*). Patients with a thicker chest wall typically have either increased subcutaneous fat or greater muscle mass, but it appears that the operator's needle insertion skill is a more significant factor for successful localization. Pneumothorax also showed as low important factor associated with dislodgement. Unlike biopsy procedures, where direct puncture of the target nodule is required and multiple needle passes or adjustments may be necessary, localization often involves marking the vicinity of the nodule without directly puncturing it. This means that localization can typically be completed with a single puncture and minimal manipulation, significantly reducing the likelihood of massive pneumothorax. Therefore, most cases result in minor pneumothorax, which is less likely to cause lung deflation or significant needle movement. For these reasons, pneumothorax is considered to have minimal relevance to localization needle dislodgement in our study.

Our study's distinctive feature lies in its analytical methodology, which combines multivariate analysis and machine learning techniques, such as SMOTE and random forest, to identify the importance of various variables, even with a relatively small and imbalanced dataset. Given the limited sample size and imbalanced nature of the data, the focus was not on developing a predictive model for dislodgement but rather on analyzing and ranking the importance of various factors influencing dislodgement. While traditional methods such as logistic regression and support vector machines (SVM) can provide indirect insights into the effects of predictors on outcomes through standardized coefficients or weights, these methods have limitations, particularly in handling categorical data. To address these limitations, we prioritized random forest and XGBoost, which are equipped with commercialized packages for determining relative variable importance (*Greenwell, Boehmke & McCarthy, 2018*). Among these, the non-linear tree-based random forest method was considered most appropriate due to its ability to intuitively handle variables with multiple categories and provide interpretable results (*Schweinberger, 2023*), XGBoost was also applied to supplement the findings. To address data imbalance, SMOTE was used in the random forest analysis, as it demonstrated better performance in AUROC calculation compared to random oversampling and undersampling methods (*Paing et al., 2018*; *Sakho, Malherbe & Scornet, 2024*).

By leveraging these methods, the study highlights not only patient-related and radiologist-dependent factors but also the significant role of hospital management factors

in hookwire dislodgement. The consistency of the top five variables identified across the two metrics (mean decrease accuracy and mean decrease Gini) and other statistical methods (XGBoost) suggests robust and meaningful findings. However, the small sample size, with only 15 dislodgement cases, presents a notable limitation, restricting the robustness of statistical analyses and machine learning models. While SMOTE was employed to address class imbalance by increasing the minority class by 600%, this approach depends heavily on synthetic data, which may distort the original data distribution and introduce biases (*Wongvorachan, He & Bulut, 2023*). This reliance on synthetic data increases the risk of overfitting, as the model may adapt to artificial patterns rather than generalizing to real-world scenarios. Furthermore, oversampling can amplify noise in the minority class, potentially inflating model performance during cross-validation. Additionally, collecting a sufficient number of dislodgement cases is inherently challenging due to the rarity of such events, which often limits the availability of large, high-quality datasets. While incorporating data from multiple institutions could theoretically address the issue of sample size, it may also introduce heterogeneity in patient populations, procedural techniques, and hospital practices, which could lead to additional biases and variability in the analysis. Additionally, structural limitations of the random forest method exist, such as its tendency to favor continuous variables in the training data and its propensity to overfit the peculiarities and noise of the training data when the sample size is small (*Paing et al., 2018*). In this study, the accuracy, sensitivity, and specificity were remarkably high, suggesting the possibility of overfitting. Thus, in this study, rather than developing a model applicable to other clinical settings, we conducted an analysis focused on reviewing past cases at our institution to identify areas for improvement and aspects requiring more attention in the medical field.

As such, future research should aim to balance the need for larger sample sizes with the potential biases arising from institutional variability, exploring multicenter collaborations while standardizing data collection and analysis protocols to ensure reliability and generalizability.

This study has several limitations. As a cross-sectional study, it cannot establish causality. As this was a single-center study with a limited sample size, generalizability of the findings may be restricted. In addition to methodological limitations, the findings of this study are more appropriately applied as a means to identify areas for improvement within the institution rather than being generalized for broader use. However, the alignment of our results with previous studies supports the reliability of our findings, highlighting the need for future large-scale multi-center studies to further validate and generalize these results. From the perspective of accuracy in data collection and coding processes, this study is the retrospective study and relied on surgical records for data on dislodgement. To mitigate this, we meticulously reviewed the surgical records and excluded cases with unclear documentation of localization success. The measured distance were measured based on axial CT rather than a 3D analysis, potentially leading to minor discrepancies. However, since most CT-guided localization procedures are performed axially, the difference is likely negligible. The procedural skills of radiologists and thoracic surgeons may have acted as confounding factors in our study. However, procedural skills are inherently difficult to quantify or objectively assess. We believe the uniformity of the procedures was likely higher

in our study, as it was conducted at a single institution where a small group of radiologists used the same methods and equipment for localization, compared to studies conducted at larger institutions. However, for surgical procedures, our study relied solely on findings recorded in operative notes, which inherently limits the scope of verification.

## CONCLUSIONS

This study analyzed factors influencing dislodgement during CT-guided hookwire localization at a single medical institution. The findings aimed to identify variables associated with dislodgement risk and determine whether any of these factors could be improved upon. The most significant variable was PS interval, followed by DNP, DWP, total depth, and age. The identification of PS interval as the most critical factor, which can be improved through system interventions within the hospital, is a key conclusion that highlights an opportunity to reduce risk through targeted improvements.

Among lesion-related factors, excluding age, variables primarily associated with nodule location—such as DNP, DWP, and total depth—showed a strong relationship with dislodgement risk. These findings suggest important considerations for planning surgeries and procedures. With these conclusions, there is potential to utilize the results for improving in-hospital medical systems and predicting dislodgement risk.

## ACKNOWLEDGEMENTS

The authors used ChatGPT (OpenAI) for English proofreading and translation assistance. The AI tool was utilized to refine the manuscript's language and ensure clarity in the translation process. Final revisions and interpretations were reviewed and confirmed by the authors.

### Funding
This work was supported by the Dongguk University, College of Medicine Research Fund of 2024. The funders had no role in study design, data collection and analysis, decision to publish, or preparation of the manuscript.

### Grant Disclosures
The following grant information was disclosed by the authors:
Dongguk University, College of Medicine Research Fund of 2024.

### Competing Interests
The authors declare there are no competing interests.

### Author Contributions

- Kiook Baek conceived and designed the experiments, performed the experiments, analyzed the data, prepared figures and/or tables, authored or reviewed drafts of the article, and approved the final draft.

- Jin Young Kim performed the experiments, prepared figures and/or tables, and approved the final draft.
- Jung Hee Hong conceived and designed the experiments, performed the experiments, analyzed the data, prepared figures and/or tables, authored or reviewed drafts of the article, and approved the final draft.

## Human Ethics

The following information was supplied relating to ethical approvals (i.e., approving body and any reference numbers):

This retrospective study was approved by the institutional review board of Dongsan medical center (IRB number DSMC IRB 2024-05-004); the requirement for patients' informed consent was waived.

## Data Availability

The data and code are available in the Supplemental Files.

## Supplemental Information

Supplemental information for this article can be found online at http://dx.doi.org/10.7717/peerj.19231#supplemental-information.

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
