# Peer review of "Analysis of factors influencing hookwire dislodgement in CT-guided hookwire localization: a retrospective study using variable importance analysis with a random forest model"

_PeerJ, doi:10.7717/peerj.19231_

## Round 0.1 · original submission · Major Revisions

Please address the detailed comments of the reviewers.

Reviewer 1 ·

Basic reporting

The manuscript entitled “Analysis of factors influencing hookwire dislodgement in CT-guided localization using a random forest model” has been investigated in detail. The manuscript investigates factors influencing hookwire dislodgement during CT-guided localization using a Random Forest model. While it tackles a clinically relevant problem, several shortcomings in methodology, reporting, and analysis undermine the manuscript's overall scientific contribution and clarity. The study demonstrates potential, but significant revisions are necessary to improve its scientific rigor and presentation.
1) With only 15 dislodgement cases, the dataset is insufficient for robust statistical analysis and machine learning, even with SMOTE oversampling. The conclusions may lack reliability due to overfitting risks.
2) The manuscript lacks proper cross-validation methods, such as k-fold validation, to confirm the reliability of the Random Forest model.
3) While SMOTE addresses class imbalance, relying heavily on synthetic data (increasing minority class cases by 600%) introduces bias, which is not adequately addressed in the discussion.
4) Parameters like tree depth and splitting criteria are not provided. A clear comparison with other machine learning models (e.g., logistic regression or SVM) is missing.
5) The manuscript identifies PS interval as the most critical factor but does not adequately explore why other significant variables like "distance to pleura" show reduced importance compared to literature findings.
6) Surgeon or radiologist skill variability is a significant confounding factor and is ignored.
7) Single-Center Study: Results from one institution limit generalizability. This limitation is not sufficiently emphasized.
8) Figures lack clear captions and proper labels, making interpretation difficult. For example, Figure 3 does not adequately explain variable importance metrics (e.g., Mean Decrease Gini vs. Accuracy).
9) The description of raw data and its preprocessing steps is inadequate, leaving replication difficult.

Experimental design

10) Expand on the knowledge gap. The manuscript should provide a stronger rationale for using machine learning over traditional statistical methods.
11) Define the scope of the research more clearly, highlighting its novelty compared to existing studies.
12) Provide more detailed information on the Random Forest model parameters, training process, and SMOTE implementation.
13) Discuss potential biases introduced by synthetic data in the analysis.
14) Incorporate a sensitivity analysis to understand the robustness of the identified factors.
15) Clarify ambiguous results, such as why variables like chest wall depth and pneumothorax rank low in importance.
16) Include more detailed statistical comparisons, such as confidence intervals for Random Forest performance metrics.
17) The authors should clearly emphasize the contribution of the study. Please note that the up-to-date of references will contribute to the up-to-date of your manuscript. The study named- “Estimation of extrusion process parameters in tire manufacturing industry using random forest classifier”- can be used to explain the methodology in the study or to indicate the contribution in the “Introduction” section.

Validity of the findings

18) Discuss the clinical relevance of the findings for surgical planning.
19) Compare results with similar studies in greater detail, particularly discrepancies in variable importance.
20) Discuss limitations more comprehensively, especially the small sample size, synthetic data dependency, and single-center scope.

Reviewer 2 ·

Basic reporting

Manuscript ID 106482v1
This paper is related to reviewing the manuscript titled " Analysis of factors influencing hookwire dislodgement in CT-guided localization using a random forest model"
This study looks into the elements that contribute to hookwire dislodgement during CT-guided localization. Methods. This retrospective research examined 123 instances of CT-guided hookwire localization with VATS resection.

Experimental design

Firstly, Although the proposed study is successful in terms of organization, presentation, content and results, major revision given in the following items need to be performed.
1) Improve the abstract and conclusion section, enhance the manuscript, especially results.
2) Use abbreviations after the first use in the text, throughout the paper.
3) The literature review is quite insufficient in the introduction section. Complete the introduction and literature sections of the manuscript by providing similar studies from the years 2023-2024 and/or new and current studies that will attract the attention of the readers.
4) Are there any other statistical and/or learning-based methods used by the authors other than the for example random forest analysis, which are well known techniques? Why are their methods weak?
---While researchers today can use artificial intelligence-based deep learning and deep learning methods to convert raw data into meaningful information and other useful results, how accurate is it to use very old mathematical methods or random forest model in this study??? Can it be said that it provides innovation for medical science? Please detail the main contribution of this study and enrich the content of the article

Validity of the findings

5) More analysis results should be included in the results and findings section. Only Figure 2 and 3 for results??
6) CT guided localization technique is not well explained, many learning models have been developed today, and the efficiency of these pure experiments is debatable.
7) The conclusion section really needs to be improved
8) Clean the paper of English spelling and punctuation errors

---

## Round 0.2 · accepted · Accept

All reviewer's queries have been adequately addressed.

Reviewer 1 ·

Basic reporting

All my comments have been thoroughly addressed. It is acceptable in the present form.

Experimental design

All my comments have been thoroughly addressed. It is acceptable in the present form.

Validity of the findings

All my comments have been thoroughly addressed. It is acceptable in the present form.

Reviewer 2 ·

Basic reporting

Accept

Experimental design

Accept

Validity of the findings

Accept

Additional comments

Accept